# Virulence and community dynamics of fungal species with vertical and horizontal transmission on a plant with multiple infections

Kai Fang[1,2◦], Jie Zhou[1◦], Lin Chen[1,2], Yu-Xuan Li[1], Ai-Ling Yang[1], Xing-Fan Dong[1], Han-Bo Zhang[1,2]*

1 State Key Laboratory for Conservation and Utilization of Bio-Resources in Yunnan, Yunnan University, Kunming, China, 2 School of Ecology and Environmental Science, Yunnan University, Kunming, China

◦ These authors contributed equally to this work.
* zhhb@ynu.edu.cn

## Abstract

The virulence evolution of multiple infections of parasites from the same species has been modeled widely in evolution theory. However, experimental studies on this topic remain scarce, particularly regarding multiple infections by different parasite species. Here, we characterized the virulence and community dynamics of fungal pathogens on the invasive plant *Ageratina adenophora* to verify the predictions made by the model. We observed that *A. adenophora* was highly susceptible to diverse foliar pathogens with mixed vertical and horizontal transmission within leaf spots. The transmission mode mainly determined the pathogen community structure at the leaf spot level. Over time, the pathogen community within a leaf spot showed decreased Shannon diversity; moreover, the vertically transmitted pathogens exhibited decreased virulence to the host *A. adenophora*, but the horizontally transmitted pathogens exhibited increased virulence to the host. Our results demonstrate that the predictions of classical models for the virulence evolution of multiple infections are still valid in a complex realistic environment and highlight the impact of transmission mode on disease epidemics of foliar fungal pathogens. We also propose that seedborne fungi play an important role in structuring the foliar pathogen community from multiple infections within a leaf spot.

## Author summary

A growing number of examples indicate that many plant diseases are caused by multiple taxa of microbes. Therefore, how virulence evolves in the context of multiple infections by different species with both vertical and horizontal transmission modes represents an important area of pathogen ecology and evolution, but there is a lack of experimental study. Here, we employ a naturally occurring host-parasite system, the invasive plant *Ageratina adenophora* and its foliar pathogens, to verify that theoretical predictions of

**Data Availability Statement:** The nucleotide sequences reported in this study have been deposited in GenBank under accession numbers MT908377-MT908827. The next-generation data

were submitted to GenBank under the Bioproject accession number PRJNA657950. Data supporting the results are attached to S1 Data.

**Funding:** HZ acquired finacially supports from the National Natural Science Foundation of China (grant nos. 31770585 and 31360153). The funders had no role in study design, data collection and analysis, decision to publish, or preparation of the manuscript.

**Competing interests:** The authors have declared that no competing interests exist.

classical models for virulence evolution are still valid in a complex realistic environment, i.e., the transmission mode determines the dynamics of the virulence and pathogen community under multiple infections. Moreover, we propose that seedborne fungi are important in structuring the foliar pathogen community consisting of multiple infections within a leaf spot. Our findings provide valuable information for understanding how multiple infections affect the key components, i.e., the virulence evolution and pathogen community dynamics, of host-pathogen interactions in the field.

## Introduction

The traditional concept of a single pathogen infecting a single host has been widely accepted in plant pathology; however, a growing number of examples indicate that multiple infections, which are caused by multiple taxa of microbes, are common for plant diseases [1,2]. Therefore, it has become increasingly important to understand how multiple infections affect key components of host-pathogen interactions, including within-plant virulence and pathogen accumulation, evolutionary trajectories of pathogen populations, and disease dynamics [2]. Classical theory predicts that virulence evolution in multiple infections depends on the pathogen transmission mode and population dynamics of the pathogen, as well as those of the host [3]. Most models assuming that parasites are transmitted horizontally (from the surrounding environment) predict that the within-host competition of multiple infections favors the increased virulence of parasites; as a consequence, the increased virulence benefits the transmission of parasites to other hosts, while the within-host competition excludes the weaker parasites and, in turn, causes pathogen diversity to decline [3,4]. If the parasites are transmitted vertically (from parent to offspring), however, natural selection benefits parasites that are less harmful to host fitness; in the case that the parasites are transmitted both vertically and horizontally, the selection for low virulence becomes stronger when host density increases [5].

Lagging behind the progress of theoretical models, however, experimental studies have focused on infections consisting of multiple strains of the same species [6]. For plant diseases, there is still a lack of experimental evolution studies on virulence evolution in the context of multiple infections by different species with mixed transmission modes, which represents the true situation of pathogens associated with any given plant host in natural systems [2,3]. Alizon, de Roode [3] proposed a three-step experimental evolution study on virulence. However, in the laboratory or greenhouse, it is difficult to perform serial passages by inoculating multiple infections of different species with distinct virulence by vertical and horizontal transmission on a single host. Finding a suitable naturally occurring host-parasite system is an alternative way to verify theoretical predictions to date.

Exotic plant introduction has been an unprecedented biogeographical experiment performed in nature, and compelling evidence indicates that evolution can occur in ecological time spans [7]; therefore, studying invasive systems may help to elucidate how a number of ecological and evolutionary processes operate at the same time scales [8]. Exotic species have frequently invaded a novel range while leaving pathogens behind [9]; however, some invasive plants in nonnative habitats inevitably encounter diverse local pathogens [10,11]. The mechanism by which invasive plants accumulate pathogens has also been partially explained in terms of ecology and evolution [10]. For example, pathogen accumulation is positively correlated with invasion time [12] and the expansion of the geographical scope of hosts [13]; high host density increases the possibility of infection [14]. Both novel associations of invasive plants with native pathogens and cointroduction of pathogens with invasive hosts are common

[15,16]. Intuitively, virulence evolution in multiple infections with vertical and horizontal transmission naturally occurs in invasive plants over time but has not been fully characterized.

*Ageratina adenophora* (Compositae), a perennial herb native to Central America, has invaded more than 40 countries worldwide in tropical to temperate regions [17]. Since the first record in China in the 1940s, the plant has been widely distributed in the provinces of Yunnan, Sichuan, Guizhou, Guangxi and Tibet, and it has continuously spread east- and northward with clear invasive history records [18]. There is evidence that *A. adenophora* can be infected by fungal pathogens in introduced ranges [17,19]. Our recent study indicates that *A. adenophora* accumulates diverse foliar pathogens from neighbors (horizontally transmitted); however, a dominant pathogen belonging to the plant pathogen-rich family Didymellaceae (Ascomycota, Dothideomycetes, Pleosporales) does not occur on surrounding native plants [20]. Thus, we hypothesize that this pathogen is co-spread with the host through seeds (and thus can be vertically transmitted) because *A. adenophora* disperses minute asexual seeds primarily by wind and water [21]. In this research, we attempted to use *A. adenophora* as a model to verify the predictions made by models of multiple infections with mixed transmission modes. We expected that within a leaf spot of *A. adenophora*: (i) multiple fungal infections with mixed vertical and horizontal transmission are common; (ii) transmission mode can impact the pathogen community structure; and (iii) the virulence and community dynamics change in a different way between the vertically and horizontally transmitted pathogens along the invasive history of the host *A. adenophora*. Here, we focused our multiple infections at the leaf spot level, rather than at the host individual level, because (i) multiple infections within a leaf spot should include both direct and indirect interactions of pathogens [2] and (ii) multiple fungal species or genotypes from one species have been observed to co-occur in the same leaf spot of *Eucalyptus* [22,23]; thus, multiple infections in one leaf spot should be common but remain to be characterized.

## Results

### Isolating and screening leaf spot fungal pathogens

A total of 2705 fungi (belonging to 451 operational taxonomic units, OTUs) were isolated from 236 leaf spots, and the disease experiment screened 1149 pathogenic fungi (266 OTUs) on *A. adenophora*, which were phylogenetically dominated by the Sordariomycete families Glomerellaceae, Xylariaceae, as well as Didymellaceae (S1 Fig and S1 Data). The proportion of pathogenic fungi varied across families; e.g., most strains from the family Didymellaceae caused lesions, but most strains from the family Glomerellaceae did not. The isolation frequencies of Pleosporaceae (Pleosporales, Dothideomycetes) and Nectriaceae (Hypocreales, Sordariomycetes) were low but contained high proportions of pathogens. At the OTU level, the common pathogens were from the genera *Allophoma*, *Alternaria*, *Colletotrichum*, *Xylaria*, and *Neofusicoccum*; in particular, the most common pathogen OTU1 (*Allophoma cylindrispora*, Didymellaceae) was at a high level in three invasive areas (Fig 1A).

In total, pathogens were successfully harvested from 212 (89.8%) leaf spots, with an average of 5.4 pathogenic isolates per spot being observed. Among these leaf spots, 164 contained > 2 pathogenic isolates from different fungal species; one leaf spot contained > 2 genotypes from a single species (Fig 1B and S1 Data). Subsequently, these 165 leaf spots, which contained 1056 pathogenic isolates, were treated as multiple infections within a leaf spot and were employed to analyze the virulence evolution and community dynamics of pathogens.

### Determining the transmission mode of *A. adenophora* foliar pathogens

We defined foliar pathogens that also occur in seeds of *A. adenophora* as seedborne foliar pathogens (SFPs); thus, SFPs infect hosts primarily by a vertical transmission mode. In contrast,

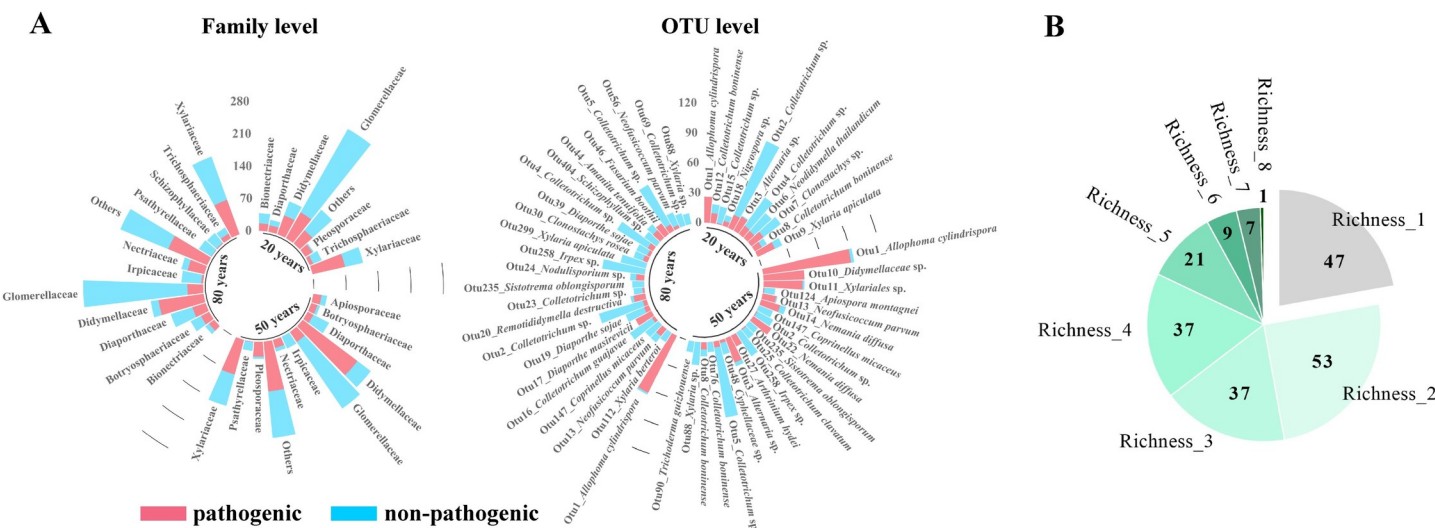

**Fig 1. Isolation and virulence verification of leaf spot fungi.** (A) The distribution of *A. adenophora* pathogens at the family (left) and OTU levels (right). Only families with relative abundance > 2% are shown, and the rest are merged into "Others"; only OTUs with relative abundance > 1% are shown. Each bar represents the abundance of one family (left) or one OTU (right) isolated from *A. adenophora* at the ~20 y, ~50 y and ~80 y sites, respectively; the part in the red represents those fungi virulent to *A. adenophora* verified by a disease experiment. (B) The categories of leaf spots with varying pathogenic fungal species. The number in each section represents the number of leaf spots with the corresponding pathogen richness. The green sections contain leaf spots with multiple infections, and the gray section contains those leaf spots with a single infection.

foliar pathogens with no exact matches to seedborne fungi were defined as nonseedborne pathogens (non-SFPs); thus, non-SFPs infect hosts primarily by a horizontal transmission mode. To determine the sharing of seedborne and foliar pathogens of *A. adenophora*, the bulked seeds of nine *A. adenophora* populations were subjected to next-generation sequencing, and 179 OTUs (591561 reads) were identified as phylogenetically belonging to 3 phyla, 27 families and 43 genera (S1 Data; for rarefaction curves, see S2 Fig). Then a comparison analysis verified that 17 fungal OTUs occurred as both seedborne and foliar fungi, accounting for 78.3% (462980 reads) and 11.5% (311 isolates) of the fungi in each pool (Fig 2A). Among these overlapping foliar fungi, 291 foliar fungal isolates were pathogenic to *A. adenophora* and were categorized as seedborne foliar pathogens (belonging to 13 OTUs, accounting for 25.3% of the total foliar pathogens); in particular, OTU1 accounted for 21.5% of seedborne fungal reads and 15.6% of foliar pathogens (Fig 2B and S1 Data).

## Diversity and structure of fungal pathogens across the host's invasive history at the macroscale and microscale level

At the macroscale level (geographic site), the Shannon diversity of the pathogen at the 80 year site was the highest (S3A Fig), and invasion time (80 years vs. 50 years vs. 20 years) had a significant effect on the pathogen community structure (S3B Fig and Table 1). However, at the microscale level (leaf spot) with multiple infections, the Shannon diversity of the pathogen decreased over time (Fig 3A). Interestingly, the fungal transmission mode (SFPs vs. non-SFPs) impacted the pathogen community structure more than the invasion time (Fig 3B and Table 1); 79 leaf spots contained SFPs, and most of them were separated from leaf spots containing only non-SFPs (Fig 3B). SFPs in leaf spots were marginally more diverse than non-SFPs, but the Shannon diversity of both decreased with invasion time (Fig 3C). When we pooled all SFP OTUs for comparison, we found that they were slightly elevated in older sites (especially at 50 years); in contrast, the non-SFP OTUs were elevated in the leaf spots of newly established populations (Fig 3D).

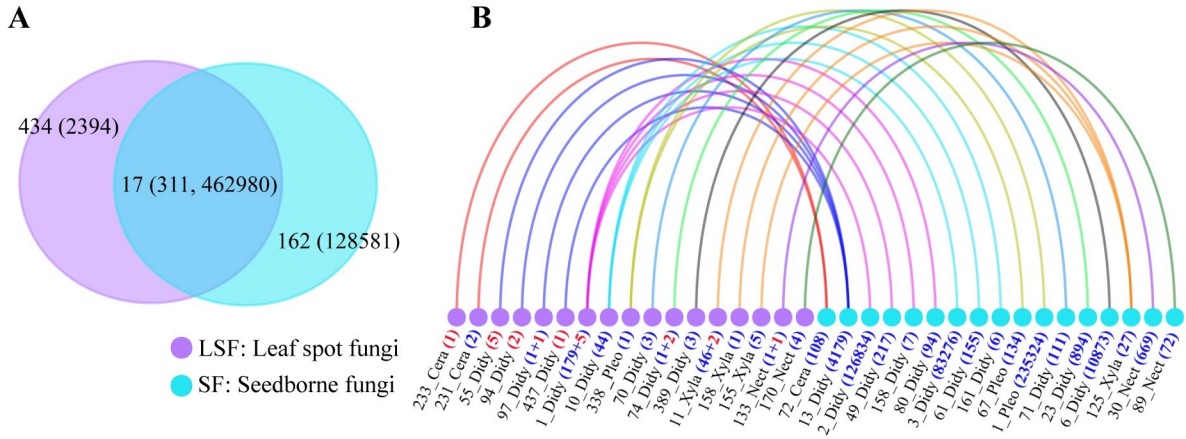

**Fig 2. Shared OTUs between leaf spot fungi and seedborne fungi.** (A) Venn diagram for the number of unique and shared OTUs between leaf spot and seed libraries. The number in parentheses represents the abundance of isolates (for leaf spot fungi, LSF) or sequence reads (for seedborne fungi, SF). The 17 shared OTUs were represented by 311 isolates of leaf spot fungi and 462980 reads of seedborne fungi. (B) Network diagram for the 17 shared OTUs between the leaf spots and the seeds. The lines depict the associations of shared OTUs between leaf spot fungi and seedborne fungi. The abbreviations following each OTU number in the corresponding library represent the family name; Didy, Didymellaceae; Xyle, Xylariales; Nect, Nectriaceae; Pleo, Pleosporaceae; Cera, Ceratobasidiaceae; the numbers in parentheses represent the abundance of isolates (for leaf spot fungi, the number of nonpathogenic fungi is shown in red) or next-generation sequencing reads (for seedborne fungi). Because the ITS sequence obtained by next-generation sequencing technology is short (~250 bp), the alignment is trimmed to this range and clustered to generate novel OTUs. This trimming causes a few OTUs to merge in both libraries, for example, OTU 1_Didy in leaf spot fungi and OTU 125_Xyla in seedborne fungi (see 4.4 Investigation of seedborne fungi of *A. adenophora*).

## Fungal virulence evolution on the host *A. adenophora*

Nearly 71% of the total pathogens were weakly virulent to the host *A. adenophora* (leaf spot area (LSA) < 50 mm$^2$) (S4 Fig). Most Glomerellaceae were weakly virulent (LSA $\leq$ 50 mm$^2$), while Didymellaceae and Xylariaceae contained the most highly virulent strains (LSA > 200 mm$^2$) (S5 Fig). The within-spot virulence variation increased linearly with pathogen diversity (Fig 4A). On average, strains were less virulent but had more variation within spots in 20-year areas than in older areas (especially 50-year areas) (Fig 4B). When analyzed separately, SFPs showed significantly higher virulence to *A. adenophora* than non-SFPs; SFP virulence to *A. adenophora* decreased, but non-SFP virulence increased, over time (Fig 4C).

## Changes in the host range and virulence to native plants

Among 184 fungal pathogens (including 28 SFPs and 156 non-SFPs) of *A. adenophora*, only 12 isolates were avirulent to any of the nine tested native species (S6A Fig). There were significantly positive correlations among fungal pathogen virulence to *A. adenophora*, virulence to

**Table 1. PERMANOVA tests for the differences in microbial community composition.**

| | Geographic site | | | Leaf spot | | |
|---|---|---|---|---|---|---|
| | **R$^2$** | **F** | ***p*** | **R$^2$** | **F** | ***p*** |
| IT | 25.193% | 1.8522 | **0.001** | 3.509% | 3.1810 | **0.001** |
| TM | / | / | / | 7.274% | 13.1894 | **0.001** |
| IT × TM | / | / | / | 1.532% | 1.3889 | **0.011** |
| Residuals | 74.807% | | | 87.686% | | |

Note: IT represents invasion time; TM represents transmission mode.

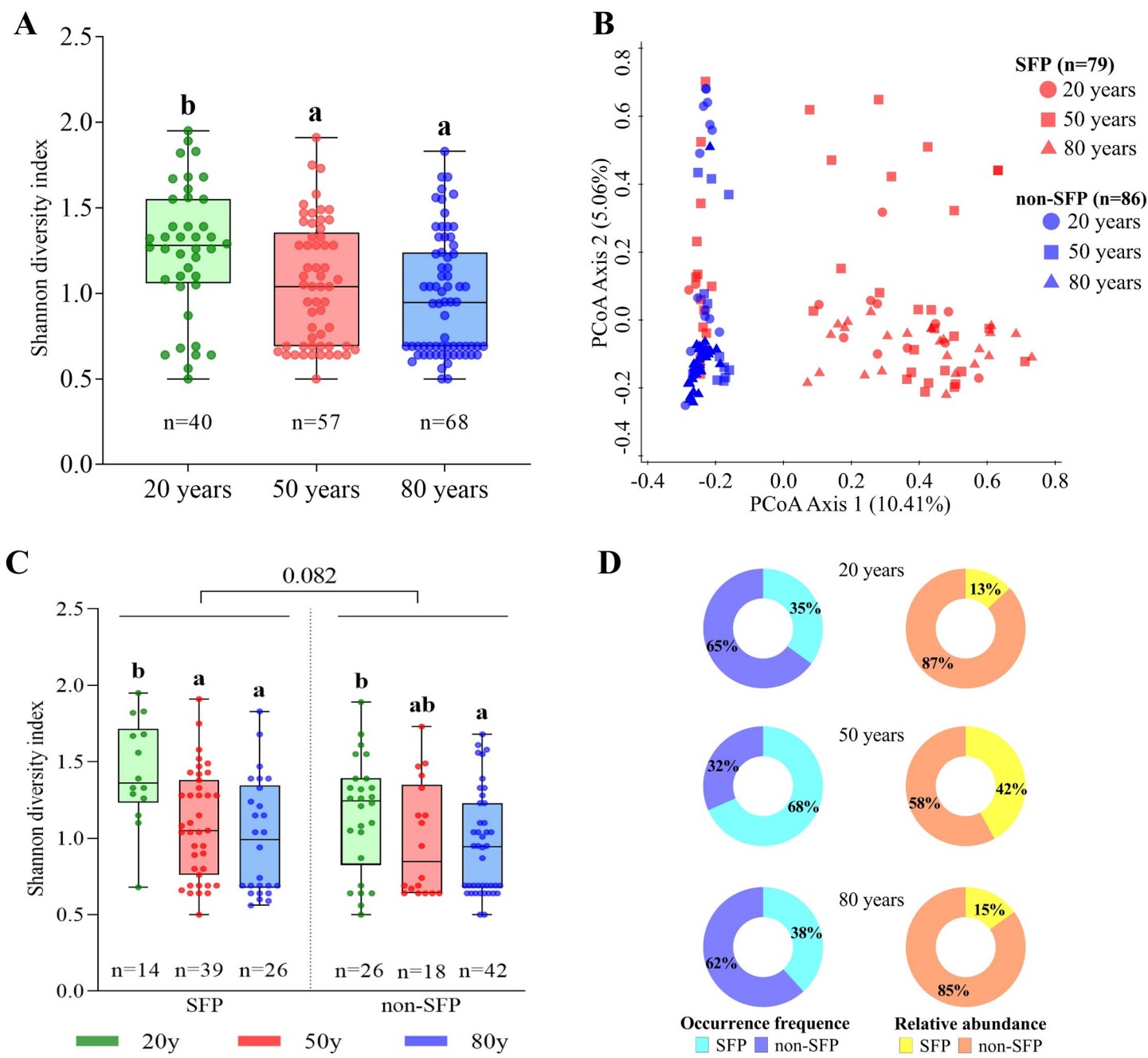

**Fig 3. Diversity and structure of the fungal pathogens within leaf spots with multiple infections over the hosts' invasion history.** Shannon diversity (A) and community structure (B) of fungal pathogens at the leaf spot level at different invasion times. (C) Shannon diversity of seedborne foliar pathogens (SFPs) and nonseedborne pathogens (non-SFPs) at the leaf spot level at different invasion times. Each point represents the pathogen community from one leaf spot with multiple infections. In panels (A) and (C), nonparametric analysis with the Mann-Whitney U test was performed to show that the Shannon diversity index difference was significant among the different invasion times based on the different lowercase letters ($p < 0.05$). In panel (B), principal coordinate analysis (PCoA) shows the similarity of pathogenic communities among leaf spots with multiple infections. SFP represents a leaf spot containing SFPs that may also cooccur with non-SFPs; non-SFP represents a leaf spot containing only non-SFPs. Percentages of the total explained variation by the PCoA axes in each plot are given in parentheses. (D) The occurrence frequency and relative abundance in leaf spots of SFPs and non-SFPs at different invasion times.

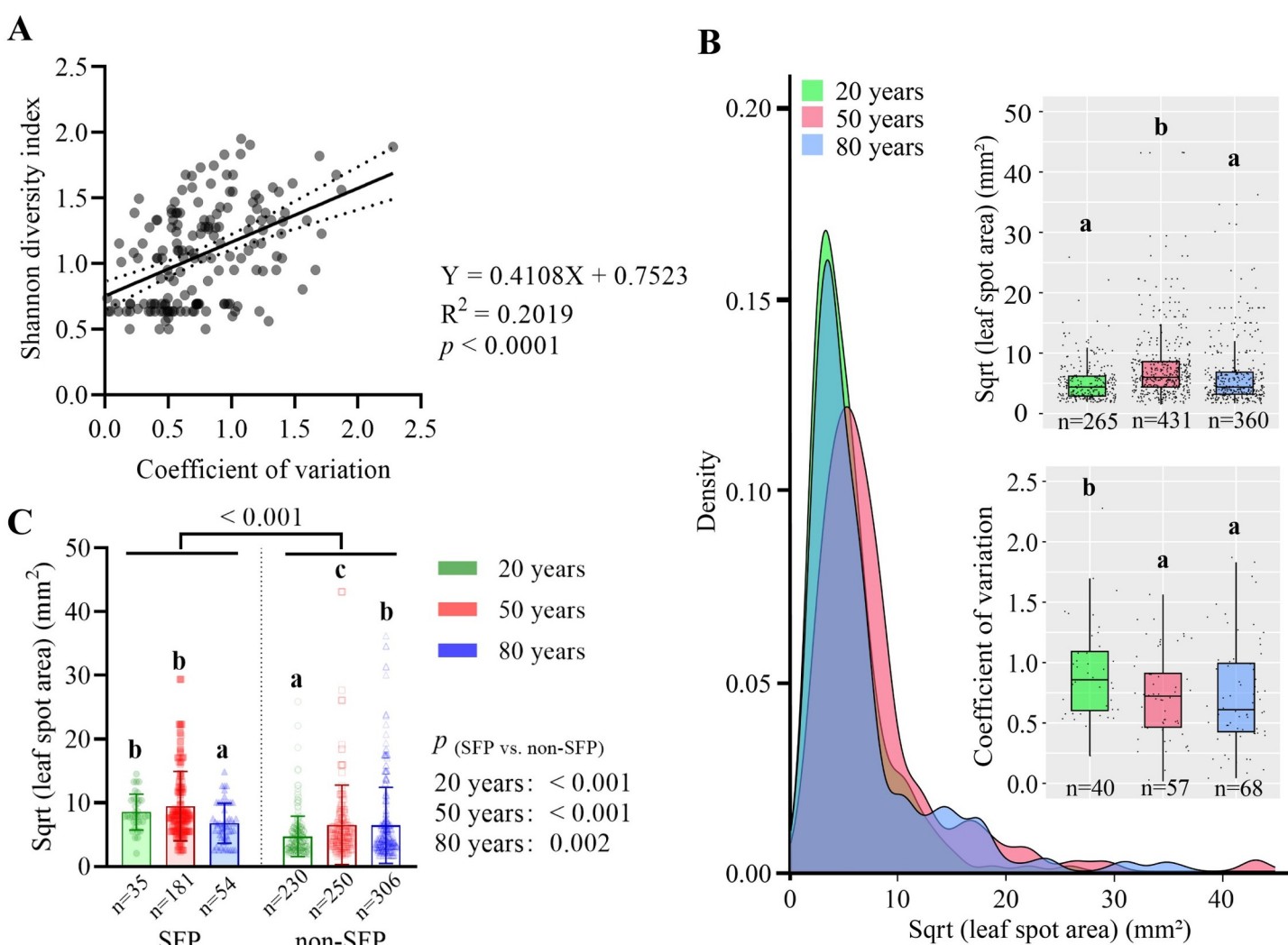

**Fig 4. Fungal virulence evolution on the host *A. adenophora*.** (A) Linear analysis of within-spot species diversity and virulence variations. Each spot represents a leaf spot with multiple infections (n = 165). (B) The density plots show the virulence distribution of fungal pathogens to *A. adenophora* at different invasion times. The boxplots show the differences in the average virulence (top) and the virulence variation (bottom) between different invasion times. (C) Virulence variation within spots of SFPs and non-SFPs at different invasion times. Nonparametric analysis with the Mann-Whitney U test was performed to test the significance of the differences, and different lowercase letters indicate significant differences ($p < 0.05$). Differences between SFPs and non-SFPs at each invasion time are also given. The leaf spot area (mm$^2$) is shown as square root transformed. The error bar in panel (C) represents the standard deviation.

native species and the pathogenic range (S6B Fig). Overall, these pathogens had no change in the host range or virulence to native plants over time (S6C Fig). In addition, both SFPs and non-SFPs were more virulent to *A. adenophora* than to native plants (Fig 5A). However, SFPs were less virulent to native plants and had a narrower host range than non-SFPs, and this pattern also did not change over time (Fig 5B and 5C).

## Discussion

Understanding how the pathogen community associated with invasive plants changes over time has been an increasing concern of biologists [12,15,16]. In general, recently introduced hosts support uniformly low pathogen richness, whereas longer-established hosts support a greater average number of pathogenic species [12]. Accordingly, we observed that invasive *A.*

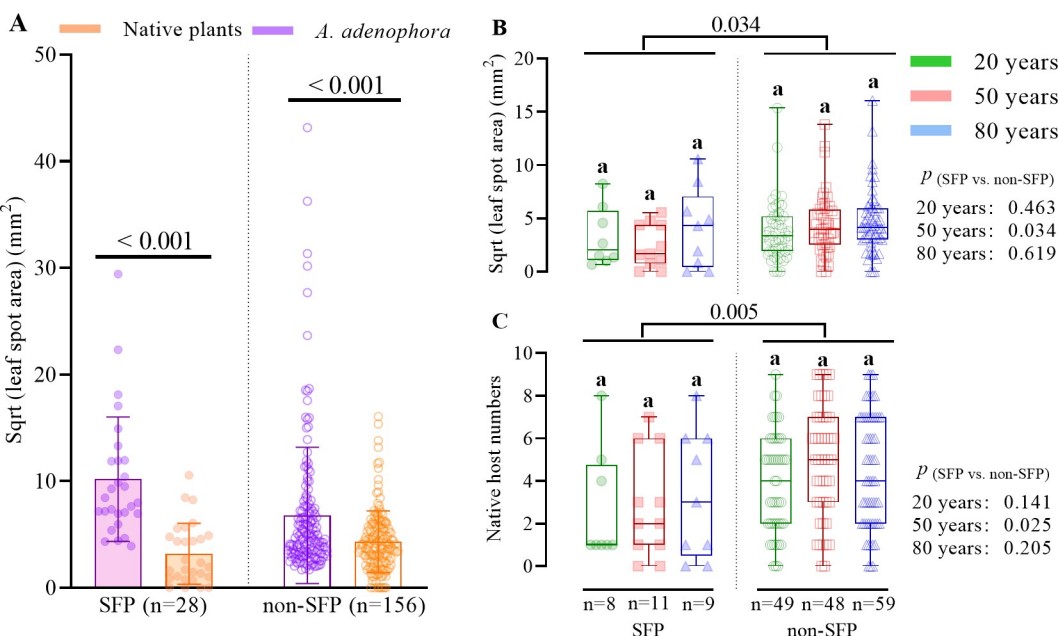

**Fig 5. The virulence and host range of SFP and non-SFP fungi.** (A) The virulence difference of either SFP or non-SFP fungi to *A. adenophora* and native plants. (B) The virulence difference and (C) host range of SFP and non-SFP fungi to native plants. Nonparametric analysis with the Mann-Whitney U test was performed to test the significance of the differences, and different lowercase letters indicate significant differences ($p < 0.05$). Differences between SFPs and non-SFPs at each invasion time are also given. The error bar in panel (A) represents the standard deviation.

*adenophora* was a highly susceptible host to phylogenetically diverse foliar pathogens in the introduced range, with the Shannon diversity increasing over the introduction time at the macroscale level (geographic site) (S3A Fig). However, our important finding was that at the microscale level (the leaf spot), 165 of the investigated leaf spots (77.8%) showed multiple infections (Fig 1B), and the change in pathogen diversity over time was reversed, with new invaders harboring more diverse pathogens in one leaf spot than older invaders (Fig 3A).

More interestingly, we found that 25.3% of foliar pathogens of *A. adenophora* were related to seedborne fungi (Fig 2A and S1 Data), which primarily determine the assembly of pathogen communities within a leaf spot (Fig 3B). The reason for this phenomenon may be relevant to the asymptomatic growth of seedborne foliar pathogens within hosts, as the primary symbiont hypothesis (PSH) indicated [24]. Our previous report indicated that members of the family Didymellaceae (the primary seedborne foliar pathogens in this case, see Fig 2) can asymptomatically grow within healthy leaves [25] and stems [26], and their growth is stimulated by leaf extracts of *A. adenophora* [25]. This manner of growth also explains why nearly 48% of leaf spots with multiple infections contain seedborne foliar pathogens (Fig 3D and S1 Data). Here, these seedborne foliar pathogens are typical necrotrophs, which can grow asymptomatically in plant tissue until becoming pathogenic under certain conditions [27,28]. Recently, a report indicated that for foliar fungi in forbs, vertical transmission from the mother plant to offspring via seeds was a widespread phenomenon [29]. Furthermore, evidence of microbial transmission routes from seeds to the phyllosphere was provided by growing oak seedlings in a microbe-free environment using an experimental culturing device [30]. Thus, it is very interesting to determine how these seedborne fungi migrate to leaf tissue, where they cause disease, and how they interact with horizontally obtained fungi during the seedling development of *A. adenophora*.

Classical theory predicts that virulence evolution in multiple infections depends on the pathogen transmission mode and population dynamics of the pathogen, as well as on the host; however, for plant diseases, there is still a lack of experimental studies [3]. Regarding the great challenge in performing serial passages in the laboratory or greenhouse by inoculating multiple different species with distinct levels of virulence, as well as with vertical and horizontal transmission on a host, we attempted to use the epidemiological investigation of *A. adenophora* and its foliar pathogens, a naturally occurring plant host–parasite system, as an alternative way to verify theoretical predictions. For horizontally transmitted pathogens, classical theoretical models regarding multiple-strain infections with varied virulence from one species predict that competition exclusion within the host favors pathogenic strains with higher virulence, which, in turn, causes lower pathogen diversity [4,31,32]. In keeping with these predictions, we observed decreased diversity (Fig 3C) and increased virulence (Fig 4C) of horizontally transmitted pathogens over time. In addition, we also observed a linear relationship between the Shannon diversity of the pathogen and the variation in virulence within spots (Fig 4A), as well as higher variation in the virulence within spots in young invasion fronts than in older invasion sites (Fig 4B). These facts imply that, in the invasion front, sporadically distributed *A. adenophora* individuals are likely coinfected by diverse native pathogens with distinct but low virulence due to life-history trade-offs when infecting novel hosts [33]; as more virulent strains evolve, they exclude weakly virulent strains and, in turn, cause decreased pathogen diversity within the leaf spot of *A. adenophora* over time. In contrast, vertically transmitted pathogens will decrease in virulence over time because their fitness primarily depends on hosts [3]; their transmission is also more effective when virulence is low [5]. The decreased virulence (Fig 4C) and the increased abundance and occurrence of seedborne foliar pathogens (SFPs) over invasion time (Figs 1A and S7) supported this idea. Interestingly, although decreased virulence to the host was observed, the virulence of these SFPs to *A. adenophora* as a whole was still higher than that of non-SFPs (Fig 4C). Because less virulent pathogens are competitively inferior in the case of mixed infections [3], it is interesting to explore whether the virulence of SFPs will constantly decrease to a level lower than that of non-SFPs or to a balance point in a certain amount of time with the delayed invasive history.

A generalist strategy provides the parasite with more opportunities for transmission and persistence; however, specialists should be favored when hosts are abundant. Moreover, the evolution of virulence also depends on the host range; in theory, decreasing specificity can be associated with increasing virulence [34]. We assume that in the invasion front, the native pathogen that can jump hosts to infect rare *A. adenophora* is most likely a generalist with a wide host range; therefore, this pathogen trends to narrow its host range with the increased host density in older geographic sites. In contrast, coinfection can offer opportunities for hybridization among different species to enlarge the host range [35]; multiple infections within a leaf spot may broaden the host range for a host-specific seedborne fungus. However, both non-SFPs and SFPs showed no change in their host range and virulence to native plants (Fig 5B and 5C). Because these pathogens changed in their virulence to the host *A. adenophora* (Fig 4C), the results implied that the virulence of foliar fungal pathogens to their associated host might be a more plastic trait than host range, regardless of the transmission mode.

Escaping, or encountering less virulent specialists if not escaping, has been considered to contribute to the successful invasion of exotic plants in the introduced range [36,37]. However, this effect is commonly significant in the early stage of invasion; with increased residence time, the pathogens accumulated by the host may directly slow and potentially stop plant invasion [10,11]. Opposite changes in virulence evolution and population dynamics for vertically and horizontally transmitted fungal pathogens in this case (Fig 4C) complicate the evaluation of pathogen impacts on plant invasion. The vertically transmitted pathogens alone seem to

suggest a declining adverse effect on the growth of *A. adenophora* over time and thus may positively contribute to *A. adenophora* expansion; in contrast, the increased virulence of the horizontally transmitted pathogens seems to slow and eventually stop plant invasion. However, our previous report indicated that foliar microbes, primarily fungi, in older sites produce stronger adverse effects on the growth of *A. adenophora* than do those in younger sites [38]. Although the horizontally transmitted fungal pathogens showed, on average, a lower virulence to *A. adenophora* than the vertically transmitted pathogens (Fig 5A), the horizontally transmitted fungi may overwhelm the vertically transmitted fungi in the growth performance of *A. adenophora*. These facts highlight the necessity to evaluate the net effect rather than that of a single species when characterizing the contribution of foliar microbes to plant invasions. Alternatively, the pathogens accumulated by exotic plants can be transmitted to co-occurring native susceptible hosts and cause a high disease risk in the invaded ecosystem [16,39]. Our data verify that most of these pathogens, particularly the horizontally transmitted ones, are virulent to multiple native plant species (Fig 5C), and their co-spread with host expansion will raise concerns about the management of emerging diseases driven by plant invasions.

Finally, our finding that the assembly of foliar pathogens in leaf spots is related to seedborne fungi may have far-reaching impacts on the understanding of what factors determine foliar pathogen community structure. Complex sets of biotic and abiotic factors have been thought to determine the foliar pathogen community [40]; in particular, host plant identification strongly affects the occurrence of foliar fungal pathogens [41,42]. Until very recently, the primary symbiont hypothesis (PSH) proposed that a single genotype (the seedborne primary symbiont) has strong effects on the microbiome assembly of seedlings [24]. In fact, the infection of a primary foliar pathogen may be common in native [43] and invasive plants [11], but it remains to be verified whether this pattern is relevant to seedborne fungi. Here, we propose that it is important to identify whether the primary foliar pathogen of a given plant host is relevant to seedborne fungi when characterizing the pathogen community containing multiple infections (mostly horizontally transmitted from the surroundings). Additionally, we emphasize studying pathogens at the genotype level [44] because identifying a given genotype, rather than one species, may be crucial to elucidating the pathogen community of multiple infections.

In conclusion, we employed the invasive plant *A. adenophora* as a model to verify many predictions of multiple infections of different fungal species with mixed vertical and horizontal transmission modes. This is the first investigation to reveal that multiple fungal infections with mixed vertical and horizontal transmission modes are common within the leaf spots of a natural plant system (the first expectation), and the vertical transmission mode determines the pathogen community structure (the second expectation). Our data also verified that the virulence and community dynamics changed in a different way between the vertically and horizontally transmitted pathogens and the invasive history of the host *A. adenophora* (the third expectation). However, our data were mainly based on epidemiological investigations and comparisons of different field populations of *A. adenophora*, without considering microhabitat differences or other potential selective factors across the sites. In addition, virulence was evaluated on host from one geographic site, ignoring the effects of host evolution. Therefore, in the future, it is still necessary to interrogate the virulence evolutionary theory regarding multiple infections with mixed transmission modes using serial passage experiments (SPEs) in a controlled environment [45]. Moreover, we categorized the foliar pathogens into vertical and horizontal transmission modes using only the sharing of seedborne fungi without verifying their within-host growth systematically one by one (see Fig 2B). In fact, these foliar pathogens may transmit by a complex mode, e.g., the seedborne OTU1 fungi can also horizontally transmit (see Chen, Fang [46]), and some non-SFPs are expected to vertically transmit if fragments of

stems and rhizomes are dispersed by the flooding during the expansion of *A. adenophora* [21]. Indeed, many human, animal and plant diseases exhibit mixed vertical and horizontal transmission modes [5] and simultaneous infection by multiple species, as well as multiple genotypes of the same species [1,2,22,23]. Nonetheless, our investigation using a large number of pathogens from different species that coinfect a spreading invasive plant could be considered a virulence evolution experiment in a realistic environment. Surprisingly, the data verify that many predictions in classical models regarding multiple infections of strains from the same species [3] remain valid in such a complex system, in which the transmission mode (vertical vs horizontal) determines the virulence evolution trajectory and community dynamics of pathogens.

## Materials and methods

### Study sites and sample collections

Previously, we reported the sharing of a foliar pathogen of *A. adenophora* with local plants in Yunnan Province, which is a relatively old invasion area [20]. In this case, we expanded the sampling areas to two newly established neighboring provinces, Guizhou and Guangxi. Because we were concerned with multiple within-leaf spot infections, previous pathogens without clear leaf spot sources were excluded from the analysis. According to the geographical expansion dynamics of *A. adenophora* in Southwest China [18], a total of 14 sites belonging to 80-, 50- and 20-year-old areas were selected for the investigation of leaf spots. At each site, leaf spots with different symptoms were collected from *A. adenophora*, and 236 leaf spots were subjected to isolation of leaf spot fungi within 12 hours after collection (Table 2 and S1 Data).

### Isolation and identification of leaf spot fungi

We focused on isolating necrotrophic pathogens because symptoms have rarely been observed on the leaves of *A. adenophora* for any of the biotrophic pathogens. The leaves were rinsed with tap water and then surface sterilized (2% sodium hypochlorite for 30 s and 75% ethanol

**Table 2. Geographical information, population distribution and number of leaf spots at three invasive times.**

| Invasion time[a] | Sample sites | Province | Longitude (°) | Latitude (°) | Elevation (m) | No. of leaf spots |
|---|---|---|---|---|---|---|
| 80 years | CY | Yunnan | 99.31 | 23.22 | 1708 | 25 |
| | LC | Yunnan | 99.80 | 22.63 | 1715 | 38 |
| | SM | Yunnan | 100.81 | 22.75 | 1171 | 22 |
| | NE | Yunnan | 101.05 | 23.00 | 1391 | 21 |
| 50 years | XS | Yunnan | 102.62 | 24.98 | 2080 | 25 |
| | YX | Yunnan | 100.03 | 24.28 | 1340 | 24 |
| | NJ | Yunnan | 100.30 | 24.87 | 1730 | 20 |
| | ES | Yunnan | 102.28 | 24.12 | 1748 | 17 |
| 20 years | NY | Guizhou | 105.58 | 26.83 | 1160 | 5 |
| | PT | Guizhou | 107.06 | 25.84 | 660 | 9 |
| | DS | Guizhou | 107.52 | 25.36 | 900 | 8 |
| | ND | Guangxi | 107.51 | 25.03 | 800 | 7 |
| | TL | Guangxi | 106.23 | 24.29 | 340 | 7 |
| | DB | Guangxi | 106.57 | 23.34 | 700 | 8 |

**Note:** [a] Invasion time for each site is from [18]. CY represents Cangyuan County, LC represents Lancang County, SM represents Simao city, NE represents Ninger County, XS represents Xishan District in Kunming city, YX represents Yunxian County, NJ represents Nanjian County, ES represents Eshan County, NY represents Nayong County, PT represents Pingtang County, DS represents Dushan County, ND represents Nandan County, TL represents Tianlin County, and DB represents Debao County.

for 2 min, followed by rinsing with sterile water 3 times). Healthy leaf tissues and the margins of diseased tissues for each leaf spot were cut into twelve ~6-mm$^2$ sections. For leaf spots < 2 mm in diameter, 3–4 leaf spots with the same symptoms from one leaf were used to represent one leaf spot. The disinfected fragments were subsequently plated onto potato dextrose agar (PDA) and incubated at ambient temperature for 6–8 d. All fungal colonies grown from the leaf fragments were purified and used in phylogenetic analysis and disease experiments. The fungi were maintained as pure cultures at Yunnan University (Kunming, China).

Total genomic DNA was extracted from fungal mycelia using the CTAB method [47]. The internal transcribed spacer (ITS) region was amplified with the fungal primer pair ITS4 and ITS5, and PCR products were sent to Sangon Biotech Co., Ltd. (Shanghai, China) for ITS sequencing. Based on the GenBank and UNITE databases, sequence homology analysis, quality assessment and correction were conducted. ClustalX v.2.1 was used to remove chimeric bases to make each sequence measure approximately 550 bp in length [48].

We grouped fungi into operational taxonomic units (OTUs) based on the ITS locus because using traditional morphological species recognition and biological species recognition of fungi is operationally difficult due to the unavailability of taxonomically informative morphologies or mating systems for many of the cultured fungi [49]. We analyzed the fungal community using OTUs based on 100% cutoffs without considering the intraspecific and intragenomic variations among ITS copies in fungi [50]. MOTHUR v.1.35.1 with the naïve Bayesian classifier was used to identify the remaining sequences (database, UNITE_public_mothur_full_04.02.2020 [51]) and to group the consensus sequences into OTUs [52]. Because we trimmed the alignment differently and used the updated version of the UNITE database in this analysis, which caused some changes to the delimitation and phylogeny of OTUs that were previously reported [20], these ITS sequences were resubmitted, and all nucleotide sequences reported in this study were deposited at GenBank under the accession numbers MT908377-MT908827 (also see S1 Data).

### Detection of the virulence and host range of leaf spot fungi

We used foliar damage as a virulence evaluation in this study. We first tested the virulence of all isolated fungi on the host *A. adenophora*. The inoculation experiment was performed as previously reported to test the virulence of necrotrophic pathogens in tropical forests [53] with minor modifications. Briefly, fungi were grown on potato dextrose agar (PDA) for 7 d, and 6 mm$^2$ agar discs with fungal mycelium were inoculated into mature and healthy leaves of *A. adenophora*. Small wounds were made by lightly touching the underside of the leaf with a sterilized toothpick, which resulted in a wound area of c. 0.2 cm$^2$. The inoculum agar was pressed against the wound on the underside of the leaf using scotch tape, which was then clipped in place (S8 Fig). Each fungal strain was inoculated into five mature and healthy leaves from a single individual of *A. adenophora* to test pathogenicity. At 1 week after inoculation, leaves were harvested and photographed, and lesions (S8B Fig) were measured. The disease experiment was performed over multiple years in the field, and environmental factors, such as moisture and temperature, were not controlled. However, all inoculation experiments were performed between June and the end of October, the primary growth season for plants in Kunming. Then, to further determine the virulence to native plants and the host range of the verified pathogens of *A. adenophora*, fungal pathogens of *A. adenophora* were randomly selected to perform the pathogenicity test on nine native plants.

### Investigation of seedborne fungi of *A. adenophora*

Because the vertical transmission of fungi in most plant seeds is imperfect and their isolation frequency is commonly low [24], we performed next-generation sequencing using bulked seed

samples rather than individual seeds. Next-generation sequencing was performed on seeds collected from nine populations of *A. adenophora* in Yunnan Province (S1 Data). We analyzed the fungal community associated with seeds using OTUs based on 97% cutoffs. Then, to identify the fungi shared in foliar and seed samples of *A. adenophora*, the representative ITS sequences of each OTU from two libraries were pooled and aligned again. The ITS sequence obtained by next-generation sequencing technology was short (~250 bp), and the alignment was trimmed to this range to cluster reads into OTUs using MOTHUR v.1.35.1 software with a 100% similarity cutoff [54]. This trimming caused several OTUs to merge in both libraries. Deletions/insertions were included when comparing sequences. The OTUs that were identical across both sources were defined as shared fungi. The shared leaf spot pathogens were defined as seedborne foliar pathogens (SFPs); in contrast, leaf spot fungal pathogens with no exact matches to seedborne fungi were defined as nonseedborne foliar pathogens (non-SFPs). The next-generation data were submitted to GenBank under the Bioproject accession number PRJNA657950 (also see S1 Data).

## Data analysis

At the macroscale level (geographic site), we pooled all fungi from each geographic site as an independent sample to calculate and compare the pathogen diversity. At the microscale level (leaf spot), only the leaf spots with multiple infections were analyzed; the pathogens in each leaf spot were treated as a pathogen community (independent sampling), and pathogen diversity was calculated and compared. Moreover, only these pathogens were used to evaluate virulence evolution and pathogen dynamics over time based on the categories of vertical and horizontal transmission modes.

Nonparametric analysis with the Mann-Whitney U test was used to compare the differences of all parameters between invasion times, including the Shannon diversity, average virulence, virulence variation and native host range of fungal pathogens. Principal coordinate analysis (PCoA) visualized the similarity in the fungal composition. All data represented fungal abundance, and distance matrices were constructed based on the Bray-Curtis dissimilarity index. Permutational analysis of variance (PERMANOVA) was used to test the differences in fungal community structure. The coefficient of variation (CV) was calculated by dividing the standard deviation by the mean to represent the virulence variation of fungal pathogens. A linear regression was employed to analyze the relationship between virulence variation and pathogen diversity.

Nonparametric analysis and linear regression were executed using SPSS v.22.0 software (SPSS Inc., Chicago, IL, USA). Ordination analysis (PCoA) was performed using CANOCO v.5.0 software [55]. PERMANOVA was conducted using the ADONIS function in the R v.3.1.2 package 'vegan' [56]. Visualization of the Shannon diversity, average virulence, virulence variation and native host range of fungal pathogens was performed using GraphPad Prism 7 (GraphPad Software Inc., San Diego, CA, USA). The remaining figures were drawn using the R v.3.1.2 package "ggplot2" [57].

## Supporting information

**S1 Fig. Relative abundance of dominant fungi at the family level.** (A) Total isolated fungi. (B) The fungi verified to be pathogenic to *A. adenophora*.
(TIF)

**S2 Fig. Rarefaction curves of next-generation sequencing for fungi in seeds.** The geographic information for the *A. adenophora* population is detailed in S1 Data.
(TIF)

**S3 Fig. Diversity (A) and structure (B) of fungal pathogens at the geographic site level**. (A) Nonparametric analysis with the Mann-Whitney U test was performed to show that the Shannon diversity index difference was significant among invasion times by different lowercase letters ($p < 0.05$). (B) Principal coordinate analysis (PCoA) shows the similarity of pathogenic fungal communities among *A. adenophora* populations. Each spot represents one geographic site (for details, see Table 2). Percentages of total explained variation by the PCoA axes in each plot are given in parentheses.
(TIF)

**S4 Fig. Virulence distribution of total fungal pathogens pathogenic to *A. adenophora*.**
(TIF)

**S5 Fig. Abundance of fungal pathogens with different virulence to *A. adenophora* at the family level (A) and OTU level (B).** (A) Families with relative abundances greater than 1% are shown, and the 14 relatively abundant families account for 86.2% of the total abundance. (B) OTUs with relative abundances greater than 1% are shown, and the 19 relatively abundant OTUs account for 50.1% of the total. The families (or OTUs) are clustered according to phylogenetic position estimated by maximum likelihood with 1000 replicates, and the bootstrap percentages are indicated at the branch node. According to the density distribution of the virulence of fungal pathogens to *A. adenophora*, 5 virulence ranges (LSA $\leq$ 50 mm$^2$, $50 <$ LSA $\leq 100$ mm$^2$, $100 <$ LSA $\leq 150$ mm$^2$, $150 <$ LSA $\leq 200$ mm$^2$, $200 <$ LSA $\leq 500$ mm$^2$ and LSA $> 500$ mm$^2$) were selected to display the data. The scale bar represents the genetic distance.
(TIF)

**S6 Fig. Host range and virulence of fungi pathogenic to native plants.** (A) Host range distribution of 184 pathogenic fungi from different invasion times. The number above each bar represents the number of isolates. (B) Pearson correlations among native host number, virulence to *A. adenophora* and average virulence to native plants of fungal pathogens (n = 184). (C) Average host range and virulence to native plants of 184 selected fungi at different invasion times (20 years: n = 57; 50 years: n = 59; 80 years: n = 68). Nonparametric analysis with the Mann-Whitney U test was performed to test the significance of the differences, and different lowercase letters indicate significant differences ($p < 0.05$). The error bar represents the standard error.
(TIF)

**S7 Fig. Occurrence frequency and relative abundance of SFPs and non-SFPs in leaf spots with a single infection at different invasion times.**
(TIF)

**S8 Fig. Detection of the virulence of leaf spot fungi on the host *A. adenophora*.** (A) Examples of inoculated individuals of *A. adenophora* in the field. (B) Examples of symptoms developed by *A. adenophora* in response to different fungal strains.
(TIF)

**S1 Data. Raw data used in this study included the leaf spot disease incidence, cultivable fungi of each leaf spot, next-generation fungal sequencing of seeds and native host data.**
(XLSX)

## Acknowledgments

The authors thank Tian Zeng, Li-Yuan Qin, Wen-Ti Zheng, and Zhi-Ping Yang at Yunnan University for help with sampling in the field and performing the disease experiment. Dr

Huan-Chong Wang and Dr Tao Xu at Yunnan University assisted with the identification of plant species.

## Author Contributions

**Conceptualization:** Han-Bo Zhang.

**Data curation:** Kai Fang, Jie Zhou.

**Formal analysis:** Kai Fang, Jie Zhou, Han-Bo Zhang.

**Funding acquisition:** Han-Bo Zhang.

**Investigation:** Kai Fang, Jie Zhou, Lin Chen, Yu-Xuan Li, Ai-Ling Yang, Xing-Fan Dong, Han-Bo Zhang.

**Methodology:** Kai Fang, Jie Zhou, Lin Chen, Yu-Xuan Li, Ai-Ling Yang, Xing-Fan Dong, Han-Bo Zhang.

**Project administration:** Han-Bo Zhang.

**Resources:** Kai Fang, Jie Zhou, Lin Chen, Yu-Xuan Li, Ai-Ling Yang, Xing-Fan Dong, Han-Bo Zhang.

**Software:** Kai Fang.

**Supervision:** Han-Bo Zhang.

**Validation:** Jie Zhou.

**Visualization:** Kai Fang.

**Writing – original draft:** Kai Fang, Jie Zhou.

**Writing – review & editing:** Han-Bo Zhang.

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
