## [Decision Letter · Decision Letter 0]

5 Apr 2021

Dear Professor Zhang,

Thank you very much for submitting your manuscript "Virulence evolution of multiple infections with vertically and horizontally transmitted fungal species in a natural plant system" for consideration at PLOS Pathogens. I must begin by apologising for the delay in coming to a decision on his submission but it has been very difficult to secure appropriate reviewers to assess this work. As with all papers reviewed by the journal, your manuscript was reviewed by members of the editorial board and by several independent reviewers. In light of the reviews (below this email), we would like to invite the resubmission of a significantly-revised version that takes into account the reviewers' comments.

As you will note from the comments by the reviewers, there is general agreement that this work addresses an interesting and important question about pathogen-host dynamics. However, the reviewers had a number of concerns about the work as presented. Reviewer 3 had a number of suggestions of possible additional analyses that would support the main contention of this manuscript. You should consider these suggestions and, if you decide not to address them experimentally, provide a clear justification as to why you don't consider them critical.  In addition, Reviewer 4 has expressed concerns regarding the lack of clarity in the statistical methodology/support for the conclusions and how this is presented in the manuscript. The reviewer provides a few examples highlighting these concerns but also points out that these concerns are not limited to those examples. The reviewer has made pertinent suggestions about how this can be resolved. Given the extent of the revisions required, we feel that a major revision is required. 

We cannot make any decision about publication until we have seen the revised manuscript and your response to the reviewers' comments. Your revised manuscript is also likely to be sent to reviewers for further evaluation.

Sincerely,

Alex Andrianopoulos

Section Editor

PLOS Pathogens

Brett Tyler

Section Editor

PLOS Pathogens

Kasturi Haldar

Editor-in-Chief

PLOS Pathogens

orcid.org/0000-0001-5065-158X

Michael Malim

Editor-in-Chief

PLOS Pathogens

orcid.org/0000-0002-7699-2064

Reviewer's Responses to Questions

**Part I - Summary**

Reviewer #1: Here the authors evaluate a theoretical evolutionary model where vertically transmitted pathogens reduce in virulence whereas horizontally transmitted pathogens increase in virulence in association with an invasive host species proving that classical prediction models are valid in a complex natural environment involving an invasive plant species.

The work done here is interesting in that it uses a naturally occurring invasion of the plant species Ageritina adenophora and its vertically and horizontally transmitted foliar fungal pathogens to set up a model to test currently held evolutionary theories. Good techniques and data analysis appear to have been used and the conclusions are justified, however, I am not convinced that the outcomes of this study are novel enough or of broad enough interest to be published in PLoS Pathogens.

Reviewer #2: This paper describes results from a study of an invasive plant Agerantina Adenophora in three sites where it was introduced 20, 50 and 80 years ago, and the community structure of its foliar pathogens. Using a combination of methods, the authors found that pathogen diversity decreased over time, while vertically and horizontally transmitted pathogens showed decreased and increased virulence, respectively. This is consistent with predictions of evolutionary models.

This is a very nice system and the authors used a variety of methods to characterize it. The data in this paper represent quite a lot of work. I like that the way that they have applied this model system to demonstrate a pattern that was predicted by evolutionary theory. This system shows promise for studies of invasive plant control, as well as plant disease dynamics in general.

Reviewer #3: This paper reports on changing virulence profiles in several fungal pathogens of an invasive plant Ageratina adenophora across several sites. The study aims to test the hypothesis that the trajectory of virulence evolution differs among horizontally and vertically transmitted pathogens. The authors find that pathogen virulence seems to evolve to higher levels among horizontally transmitted fugnal OTUs while attenuates in vertically transmitted (or, more accurately, fungi found in both seeds and leaf spots). On the whole I think the results are suggestive and informative.

I am hard pressed to characterize the results as an "experimental evolution" study. Such studies begin with the ancestral genotype and through controlled conditions experimentally evolve organisms over several generations. The authors claim that the invasion process provides such a "natural experiment" but neither I nor any one else involved in experimental evolution will be much convinced this description is really an experimental test of how the mode of transmission affects virulence evolution. If the authors inoculated these plants over multiple generations in a green house with different kinds of pathogens and compared how the pathogens evolved during the course of that evolutionary experiment that would be one thing. This is fundamentally a different kind of test.

This is at heart an observational study using comparisons of different field populations. I think such studies can be immensely powerful and can help interrogate evolutionary theory in interesting ways, but their limitations are very well known. But the authors for instance cannot control for microhabitat differences, different host pools and other potential selective factors across the sites in Table 1. I think the use of a large number of OTUs helps mitigate this issue, but these constraints on occasion could be better emphasized.

Reviewer #4: I was asked to review the submission by Fang and colleagues with a specific focus on the statistical method used and the reporting of the data analysis. The analysis has clear merit and the main message is clearly of interest to a broad community of both fungi and plant biologist. Unfortunately there is a few points that need reworking to allow a thorough understanding of the study outcomes and potential replication of the analysis.

**Part II – Major Issues: Key Experiments Required for Acceptance**

Reviewer #1: (No Response)

Reviewer #2: None.

Reviewer #3: Virulence is both a property of host evolution and pathogen evolution. The use of a single host to test virulence (before comparing to native plants) helps control for this, but it would be important to know where this host came from and how the virulence profiles among fungi differ across host lineages (as opposed to host species). Where the pathogen's profile sits on its current host seems to be just as important in confirming or testing hypotheses about virulence evolution. For instance, a pathogen can be avirulent towards its original host, but may remain virulent on a derived host if the host's susceptibility to adverse consequences of infection has also increased. I am not particularly convinced that the consideration of non-native plants suffices as a substitute for this.

Thus I think the authors should test virulence patterns on ideally hosts at multiple invasive populations and hosts at different extremes (20 yrs. v. 80 yrs env.) to ascertain the effect of hosts as well as pathogens (so that virulence evolution for instance actually is attenuated)

Relatedly, I would like to see the community-structure account for source plant: it seems logical to me that pathogen communities from a leaf spot on the same plant are likely to show some structure. This can be done using a mixed-model with source plant as a random effect.

Reviewer #4: 1/ In the present submission, the order of introduction of the statistical methods is not consistent with presentation of the results. For example the use of a linera regression and the resulting test using an F statistic comes later than the description of the Mann-Whithney test. Make sure you are mapping rigorously the method description against the results for an easy understanding.

2/ Althogether, most test used are quite cononical and well ackowledged and don't need excessive description in the materials and methods. However, when presenting a test statistic, make sure to use a specific and rigorous terminology. As an example when you describe in line 143 "pathogen communities were clustered by fungal transmission", you should write that "fungal transmission mode (seedborne pathogens (SPs) vs. non-seedborne pathogens (non-SPs) had a significant effect on pathogen community abundance? diversity/Shannon index?". Be precise about which exact dependant and independant variables you are using in the test, and make sure that the presentation of the results eaclty matches the labels in the figures.

3/ Use a more appropriate term than % explanatory variables which I have not found elsewhere: you can say this variable explained X% of the variation for the shannon index (R2=Y, adj-R2=Z). Also report to figures as (Fig. S6b) for examples not (S6 Fig, panel b).

**Part III – Minor Issues: Editorial and Data Presentation Modifications**

Reviewer #1: The authors should do a better job of clearly defining terms, at their first use, for a general audience. These terms include multiple infections, horizontal and vertical transmission, and OTUs to name a few.

Overall, the figures are difficult to interpret. More descriptive figure legends would be helpful.

Reviewer #2: I have two general critical comments:

1. The authors discuss evolutionary theory in the Introduction, and then return to it in the Conclusions, but a lot would be clarified by a figure that lists underlying hypotheses, the predictions, and results that either support reject. At the very least, set the paper by enumerating hypotheses and/or predictions in the Intro, and then explicitly refer back to them in the Conclusions.

2. I think there are some issues with how this paper deals simplistically with the definition of “pathogen.” I think it is appropriate for this paper, but those shortcomings should be discussed. The authors performed detached leaf assays as a means to define virulence. This is fine, and it is done in many studies, but of course it is a limited tool. An isolate can cause lesions on a detached leaf and be of no consequence in an actual ecosystem. The authors mention also that many of the pathogens are known to grow asymptomatically in plants. It is well established that many, perhaps a majority, of plant pathogens can grow asymptomatically in plants, and thus fit a certain definition of ‘endophyte.’ We also know that such plant pathogenicity is often determined by “accessory” elements of the genome, that are potentially horizontally transmitted. It is possible that non-pathogenic and pathogenic forms of the same taxon may harbor a leaf spot, and that the non-pathogen could be converted to pathogen through some form of transfer. I do not believe that this study needs to deal directly with these issues, but it should acknowledge (add a paragraph to the Conclusions) that they have an effect on the results shown. I still believe the results support the conclusions.

I made a number of minor editorial suggestions on the Word document attached.

Reviewer #3: Line 32-34: I did not understand what the authors mean here and found the wording confusing.

Line 39: check tense "occurring"

Lines 40-48 seem to just repeat the abstract.

Line 80: "ecological time"

Line 82: "same time scales"

Line 155-156: the higher virulence variation is an interesting result but I couldn't find much in the discussion about it.

Line 159-167, also methods 320-332: I never really understood what the role of the host range analysis in this paper was. It would seem a better test would be to look at different host strains/populations within the same lineage. See my summary/major comment.

Lines 176, 195, 235: typo

Line 185-187: I think it would help the authors to remind the reader where this is in the results

Line 202-203: I'm not really sure this conclusion is warranted.

Line 220-223: this feels too speculative.

Lines 238-240: I think this illustrates the role of demography nicely, but there is little discussion of that in the paper

Lines 343-345: I confess to not being an expert in fungal pathogens of plants. Do seed-borne pathogens necessarily go on to infect their host and form leaf spots? How predictive is being found in the seed predictive of a pathogen being found in the same individual at a later developmental stage? How do we know this pathogen wasn't horizontally transmitted from another individual who happened to get it vertically? I think a short sentence or two giving the basics of these sorts of issues will reassure readers working in other systems of the appropriateness of the SP/non-SP classification.

Reviewer #4: You do not necessarily need to describe the libraries you used to draw the graphs and plots, these are not the critical part of the analysis. However, be mindful of introducing the sample sizes when making comparison, either through the reporting of degrees of freedoms when reporting a statistical test, or putting n=100 on the figure labels.

Finally, I do think the manuscript would benefit from a professional English editing service to make sure there is no miscommunication. For example when you use the verb "overlook" in line 312, you mean the exact contrary.

PLOS authors have the option to publish the peer review history of their article (what does this mean?). If published, this will include your full peer review and any attached files.

Reviewer #1: No

Reviewer #2: No

Reviewer #3: No

Reviewer #4: No
---

## [Decision Letter · Decision Letter 1]

29 Jun 2021

Dear Professor Zhang,

We are pleased to inform you that your manuscript 'Virulence and community dynamics of fungal species with vertical and horizontal transmission on a plant with multiple infections' has been provisionally accepted for publication in PLOS Pathogens.

Best regards,

Alex Andrianopoulos

Section Editor

PLOS Pathogens

Brett Tyler

Section Editor

PLOS Pathogens

Kasturi Haldar

Editor-in-Chief

PLOS Pathogens

orcid.org/0000-0001-5065-158X

Michael Malim

Editor-in-Chief

PLOS Pathogens

orcid.org/0000-0002-7699-2064

Reviewer Comments (if any, and for reference):

Reviewer's Responses to Questions

**Part I - Summary**

Reviewer #1: (No Response)

Reviewer #2: I believe that the authors adequately responded to my feedback on the manuscript.

Reviewer #3: On the whole the authors have addressed my key concerns.

**Part II – Major Issues: Key Experiments Required for Acceptance**

Reviewer #1: None

Reviewer #2: I am satisfied with the author's revision in response to my feedback.

Reviewer #3: The responses are a bit superficial, but not being an expert in plants/plant fungal pathogens I am fine deferring to the editor and other reviewers about how satisfactorially the details of the system are adressed.

**Part III – Minor Issues: Editorial and Data Presentation Modifications**

Reviewer #1: (No Response)

Reviewer #2: I am satisfied with the author's revision in response to my feedback.

Reviewer #3: (No Response)

PLOS authors have the option to publish the peer review history of their article (what does this mean?). If published, this will include your full peer review and any attached files.

Reviewer #1: No

Reviewer #2: No

Reviewer #3: No

---

## [Editor Report · Acceptance letter]

9 Jul 2021

Dear Professor Zhang,

We are delighted to inform you that your manuscript, " Virulence and community dynamics of fungal species with vertical and horizontal transmission on a plant with multiple infections ," has been formally accepted for publication in PLOS Pathogens.

Best regards,

Kasturi Haldar

Editor-in-Chief

PLOS Pathogens

orcid.org/0000-0001-5065-158X

Michael Malim

Editor-in-Chief

PLOS Pathogens

orcid.org/0000-0002-7699-2064